# Economic Diversification Potential in the Rentier States towards a Sustainable Development: A Theoretical Model

**Abdullah Kaya [1],\*, Evren Tok [2], Muammer Koc [1] , Toufic Mezher [3] and I-Tsung Tsai [4]**

[1]  Engineering, Hamad Bin Khalifa University, Doha, Qatar; mkoc@hbku.edu.qa
[2]  College of Islamic Studies, Hamad Bin Khalifa University, Doha, Qatar; etok@hbku.edu.qa
[3]  Engineering, Khalifa University, Abu Dhabi 54224, UAE; toufic.mezher@ku.ac.ae
[4]  School of Economics and Management, Tongji University, Shanghai 200092, China; itsung@tongji.edu.cn
\*  Correspondence: akaya@hbku.edu.qa

**Abstract:** This paper develops a theoretical model to analyze whether a rentier state can diversify its economy away from the rent revenue and hence sustain the economic development and preserve the status-quo. Considering the decarbonization process of the global economy and rapidly fall in economic value of hydrocarbons in the face of the supply glut, rentier states depending on oil and gas revenues urgently need to diversify their economies to avoid social backlash and political upheaval. There are three intertwining factors that determine an effective economic diversification away from the rent revenue: The profitability of non-rentier sectors, the size of the domestic economy to induce a "Big Push" for industrialization to non-rentier sectors, and the level of economic inclusivity. For an optimal level of economic diversification in a rentier state: (1) Non-rentier sectors should be attractive to private agents without the entry barriers; (2) domestic economy should be large enough to induce investment into non-rentier sectors; (3) the ruler(s) should have sufficient tolerance (inclusivity) for private agents investing into non-rentier sectors. Our findings indicate that a rentier state can achieve an optimal level of economic diversification provided that the conditions above are met even without any political change.

**Keywords:** rentier state; economic diversification; theoretical model

**JEL Classification:** D72; E02; O10; O14; P16; P48; Q32

## 1. Introduction

There is a significant number of countries whose economies depend on the export of commodities, such as oil and gas, minerals and other natural resources. Revenues from natural resources have propelled those countries to rapid economic development and very high living standards [1]. Many of those developing countries, however, possessed with natural resources have failed to develop a significant industrial base and diversified economy [2]. This dependence on revenue from natural resources causes sharp ups and downs in the economy and poses a significant risk for a sustainable political and economic system in the medium to long term. Many countries have experienced erosion of their industrial base after the discovery of natural resources, a process which touted as Dutch Disease [3]. Countries which are deprived of strong state institutions have witnessed civil strife and complete chaos in the face of a big natural discovery instead of prosperity a dichotomy being dubbed as Resource Curse [4].

Among those countries, a sizeable portion has an absolutist political system with considerably limited participation of general populace in the political life [5]. There are numerous theories in the

literature of economic development in analyzing whether there is a causal relationship between a rentier economic structure and an absolutist political system [6]. The Rentier State Theory (RST) has been proposed to explain the unique economic-politic nature of the authoritarian and resource-rich countries starting in the 1970s [7,8]. In a rentier state, a small segment of the society has access directly in the creation of wealth while the rest involves in distribution and utilization of the wealth [6,8]. The "rentier social contract" (rentier bargain, ruling bargain) is one where the rentier government distributes accrued wealth to society through services, social benefit programs, and favorable governmental jobs in exchange for the society's refrain from obtaining political power [9,10]. The wealth is usually accrued through extraction and selling of valuable natural resources under the full control of the government [11], which is mainly directed by a ruling elite group. The major task of a rentier state is, therefore, the distribution of wealth as opposed to extracting rent from the population in any form of taxation [11]. All countries in the Arabian Gulf, such as Bahrain, Kuwait, Oman, Qatar, UAE and Saudi Arabia, which all form the GCC (Gulf Cooperation Council), are generally categorized as rentier states where political authoritarianism has been compensated and balanced with the distribution of rent revenues from oil and gas sales to the international investments [12]. Apart from the Arabian Gulf States, Venezuela, Gabon, Russia, and Algeria are also either completely or quasi-rentier states where oil and gas exports dominate the economy with an absolutist political system [13].

Amid the urgency of the fight against climate change, there have been concerted efforts on a global scale to decarbonize the energy and transportation systems with clean and renewable substitutes [14]. Exponential and rapid advancements in solar energy, wind energy, and electric vehicle technologies pose a significant threat to the future demand for oil and gas and hence to their monetary value [15]. The discovery of new oil and gas resources (shale oil, tar sands, deep water reserves) coupled with a weakening global demand for these fuels result in entrenched supply glut and pose a significant challenge to the dominance of fossil fuels in the energy markets and to the revenues of to the rentier states [16,17]. International oil prices have fallen down more than 50% in less than six months in 2014, due to that supply glut in the global oil market [14]. Furthermore, steep fluctuations in oil and gas prices are historically normal and a detriment to economic growth and stability of rentier states [18,19]. Indeed, volatility in the oil and gas markets go beyond economic stability and growth which threatens the very delicate political and social balance as in such rentier states as described above. Increasing unemployment, especially among the young population, and rapidly deteriorating social services, due to insufficient spending are becoming stumbling blocks for sustainable development in many of those rentier states [20].

Rentier states are often acquainted with these challenges and have been developing and introducing various economic diversification programs as a response [1]. Many of the governments in those states are actively seeking to diversify their economies away from over-reliance on rent revenue in a bid to avoid economic downturn and social upheaval without compromising on their control of the state and its affairs [21]. Albeit some local and temporal successes, the economy in many rentier states is still heavily dependent on oil and gas revenues with idiosyncratic results from diversification schemes [22,23]. Therefore, economic diversification to lessen reliance on rent revenues is an urgent issue to all rentier states in the world to achieve economic prosperity and preserve political and social stability [24]. Considering the recent chaos and turmoil in some developing countries, such as Egypt, Syria, and Libya, due to collapse in the economy and state authority, any economic diversification plan should take into account the centrality and pursuance of its interests by a rentier state for being effectively implemented [10,25]. The economic success of countries with a similar political system, such as China, Singapore and the city of Dubai indicate that the rentier states may achieve a similar economic development and lessening the reliance on the rent revenue [26–28]. However, this dual objective of preserving political status-quo with economic diversification is a daunting and highly challenging task considering the fact that most economically advanced countries have inclusive political systems [29].

The theoretical model proposed in this paper lays out a fundamental framework for the potential of effective economic diversification in the rentier states under the current political system and its

continuance. It is imperative to consider economic activities in a rentier economic system with its political implications as both are interlinked with each other [29]. The current literature about rentier states and development studies tend to ignore political repercussions of economic diversification schemes in rentier states [6,12]. Any economic diversification proposal cannot be separated from its political consequences, which will determine its effectiveness at the end [30]. This paper aims to fill this gap by simultaneously taking political and economic factors into account in facilitating effective economic diversification of a rentier state. There is currently no theoretical model being proposed to explain economic diversification in a rentier state amid the flurry of reports and papers highlighting the urgency and necessity of such an economic diversification [6,10,12]. The model proposed in this paper aims to identify factors for effective economic diversification in a rentier state in the face of declining rent revenues by considering:

- The current political system as unchanging [6].
- Decreasing revenue from natural resources as inevitable [15].

Economic diversification is referred to as the substitution of imported tradable goods and services with domestically produced tradable goods and services within non-rentier sectors, such as [31]:

- Manufacturing goods of cars, home appliances, electronics, textiles, industrial equipment, and machinery, etc.
- Information and technology services, technical consultancy.

An effective economic diversification in those states should promote the expansion of tradable sectors. Domestic production of tradable products is expected to yield value-added goods and services while creating high-paying jobs for the citizens [3]. Domestic production of tradable goods and services will decrease the non-energy current account deficit in these states [21]. Therefore, it is highly assumed that decision-makers in these states would like to promote an economic diversification scheme if their core interests, such as political and social stability, are not threatened [32]. The proposed model in this paper takes all these factors into account and optimizes an economic diversification plan without a political change or upheaval as any such scenario would be rejected by the authority. The policy suggestions from the model are expected to create a sustainable economic structure in a rentier state in terms of employment, decreased dependence on rent revenues and diversified economy in terms of manufacturing and services. Considering the fact that many rentier states are classified as emerging economies with very high levels of young and growing population, sustaining economic and political order has become critical more than ever, due to rapidly changing global dynamics [33,34].

The basics of the theoretical model, including main assumptions, decision makers, and parameters are presented in Section 2. The potential scenarios and outcomes of the model are then discussed in Section 3. Conclusion and future work are summarized in Section 4.

## 2. A Theoretical Model for an Effective Economic Diversification in a Rentier State

The paper analyzes how a rentier state can diversify its economy towards tradable (non-rentier) sectors once rent revenues start declining while the political status-quo remaining the same. The economic diversification is referred to the substitution of imported tradable goods and services with domestically produced ones by domestic sectors denoted as non-rentier. There is no significant manufacturing of capital goods used for production (machinery and heavy equipment) in rentier states [17]. This has resulted in the imports of necessary capital goods from developed countries. Demand for basic consumer goods, such as food products, automobiles, home appliances, and apparels are also imported from foreign countries in most cases [35]. The state gives exclusive licenses to the family businesses established by the citizens to perform importing of some capital goods and most of the consumer goods as part of the rentier agreement.

The ruler (government) is the only decision maker and political authority that distributes rent revenues to the population in exchange for political acquiesce. To stay relevant and protect its political

power, the ruler will prevent the private agents from amassing too much economic power in many cases by partnering and obtaining a significant share instead of absolute prevention. The private agents will invest in either rentier or non-rentier sectors based on expected profitability and the ruler's approval. If profits in non-rentier (tradable) sectors are too low, then no agent would like to invest in these sectors resulting in an ineffective economic diversification. Another important group in a rentier state political and economic system is the population who provides labor, consumption and recognition to both the ruler (public sector) and the private agents for both sectors. The population has an implicit agreement with the ruler for high paying public jobs in return for giving up on political rights. The population does not prefer to work in low paying jobs of nontradable sectors (rentier) that are mostly occupied by migrant labor. However, a part of the population may prefer to work in tradable sectors, due to the premium wages offered.

Once the rent revenues fall, the ruler will face a decision on the level of economic diversification. They have a primary objective of protecting his political, social and economic power. Their secondary objective is to promote economic diversification by granting business licenses to the private agents for tradable sectors. On the other hand, the private agents move with the motivation of profits when deciding whether to move into tradable vs. nontradable sectors after they get the licenses from the ruler. There are two important factors which are (1) ruler's tolerance to economic diversification level and (2) the private agents' profitability prospect in tradable sectors. Dynamics of these two factors' interaction will be decisive for an eventual number of agents in tradable sectors. The degree of economic diversification in a rentier state is measured by the number and prospects of the private agents in the tradable sectors.

The paper first introduces the basic economic structure of a rentier state, including the main decision makers and economic sectors. This is then followed by a list of assumptions which were partially adopted from the different theories in development economics, such as The Dutch Disease Theory, The Theory of Institutions, and The Big Push Theory regarding the price movements, sectoral wages, and level of value creation. The decision variables, exogenous shock and potential scenarios as a result of these new dynamics in the following period are explained. Finally, the solution of the model is presented to come up with optimal policy decisions for effective economic diversification.

*2.1. The Model Basics*

In this proposed model, a partially open economy is assumed, which is divided into two main sectors as tradable and nontradable following the Dutch Disease literature [36]. Additional assumptions are as follows:

- The internationally tradable sector is assumed to be comprising manufacturing goods, such as steel, clothes, textiles, machinery, engines, transport equipment and durable consumer goods, and the specific services, such as consultancy, IT and logistics.
- Non-tradable sectors are those, such as general services (banking, telecommunications, and healthcare), recreational businesses, and construction activities.
- While the price of a tradable good is set in the international market, the price of the non-tradable good is set in the domestic economy.
- Production characteristic of tradable sectors has increasing returns to scale (IRS), but requires a significant upfront capital investment (fixed cost) and skilled labor, which demand premium wages.
- The private agents will not enter into tradable sectors if there is no sufficient domestic demand to cover fixed cost and make a profit.
- Non-tradable sectors require much less capital investment and have constant returns to scale (CRS) production function.

The economy of a rentier state has three main actors, which are (1) the ruler as the only decision maker, (2) the private agents investing into tradable and nontradable sectors, and (3) the rest of the

population who provides the labor, consumption and recognition. The following are the main roles and actions assumed by the actors, and Figure 1 illustrates these dynamics in a rentier economy:

- The ruler allows extraction, production, transportation and sales of natural resources to international and/or joint ventures and/or national corporations at varying levels at different times balancing the power and economic returns seeking recognition and acceptance from his population; and seeking recognition and protection from his international partners or allies.
- The ruler precedes the distribution of rent revenue (*R*) from sales to the population through public employment in return for their political acquiescence (rentier agreement).
- Therefore, rent revenue (*R*) is a significant source of political power and driver of domestic consumption.
- The ruler owns a significant share in the non-tradable sectors (banking, healthcare) to control a significant portion of the economy to prevent private agents from getting too much economic power at his expense.
- In the case of declining rent revenues, the ruler will face a dilemma of decreasing public wages or lay off some employees.
- Both of these options are undesirable in the rentier, such as GCC, states since they may induce challenge to the political authority of the ruler from the population [37].
- Hence, the ruler will support the diversification of the economy towards tradable sectors in order to create good paying jobs for the citizens as long as his share of the economy is not compromised and his political and social authority lasts.

This economic model is a two-period event where, in the first period, the events above take place under normal conditions. In the second period ($R \rightarrow R/2$), an exogenous shock to commodity prices cause a decrease in rent revenues. This forces the rulers to seek for economic diversification to avoid social backlash and chaos. Then, the ruler decides how many business licenses to be distributed to the agents for non-tradable and tradable sectors to optimize economic diversification with his share in the economy above a certain level. The private agents will choose between two sectors until profits per agent become equal in both sectors or the maximum number of agents allowed in that sector by the ruler has been reached. To simplify the model, there is no saving or investment in both periods with the exception of fixed cost for tradable sectors for both periods. All the rent revenue (*R*), profits of private agents, and wage labor are consumed in the same period for tradable or non-tradable goods.

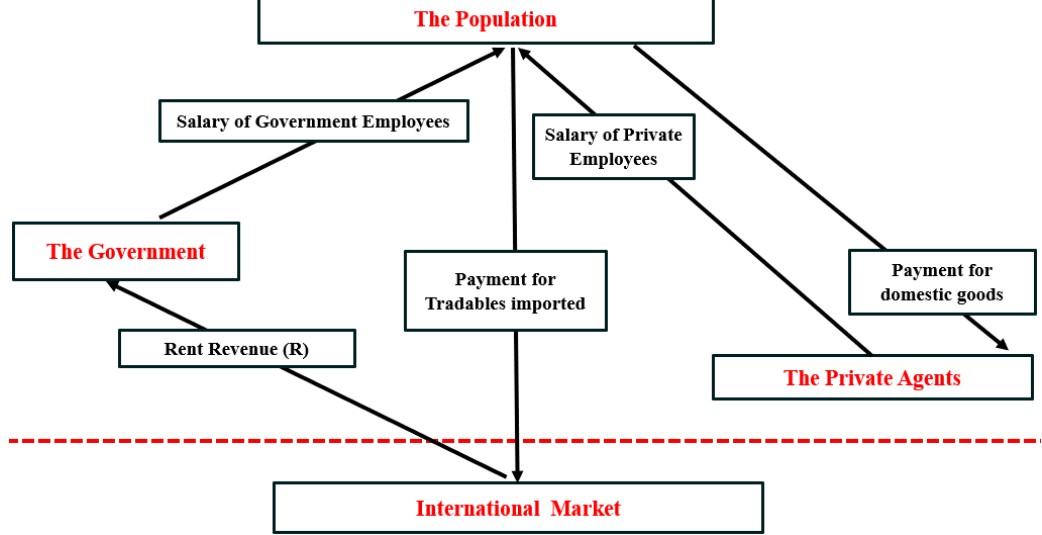

**Figure 1.** Economy dynamics of a Rentier States.

## 2.2. Assumptions of the Model

A1: In tradable sectors, there are *K* tradable sub-sectors in total where $N_T$ of them are produced locally and the rest $(K - N_T)$ is imported, in which $N_T$ represents a total number of tradable agents in the domestic economy.

A2: The price of a tradable good regardless of its sector is set in the international market as 1 (numeraire). Local producers are thus price-takers and cannot charge more than 1, which will result in import of the good. We assume that there will be no tax imposed by the ruler.

A3: Following A1, there are $N_T$ agents focused on the production of tradable goods where $N_{Tmax}$ < *K*/2. This states that no country can specialize in the complete production of tradable goods.

A4: Each tradable sector is controlled by one firm/agent and hence there is a local monopoly on the production of this good. It is assumed that quality assurance is provided by the government to prevent domestic production and distribution of low-quality tradable goods.

A5: There are increasing returns to scale (IRS) in the production of tradable goods and it is as follows:

*$X_T = \beta \times l_T - F$ where $X_T$ is the amount produced, $l_T$ is the amount of labor used with $\beta > 1$ due to IRS or using advanced technology (opposed to fringe production). Here $\beta$ is a parameter and represents the technological advancement of the economy/country meaning that the higher country's advancement, then higher $\beta$. F represents initial (fixed) investment cost measured as labor units according to Murphy et al. (1989). In that equation, F has the same unit with labor and output.*

A6: There are constant returns to scale (CRS) in the production of a nontradable good and it follows:

*$XN = lN$, where lN is the amount of labor used. We assume no fixed capital costs for the production of non-tradable goods or services.*

A7: The price of non-tradable goods (*Pn*) is set in a local market based on demand and price elasticity with respect to tradable goods and services. The price of tradable (*Pt*) can be assumed as the real exchange rate similar to previous studies [37]. Increasing rent revenue ®and its spending on the domestic economy will lead to an increase in *Pn* and profits of agents in non-tradable sectors which will increase the production of non-tradable. Accordingly, labor demand (*Ln*) in nontradable sectors will increase proportionally with prices of nontradable goods and services (*Pn*). There is appreciation in local currency and increase in the price of nontradable (*Pn*) as in Figure 2 where nontradable market goods scenario (NTGME) move from point A to A *, due to the influx of more rent revenue to the economy [29].

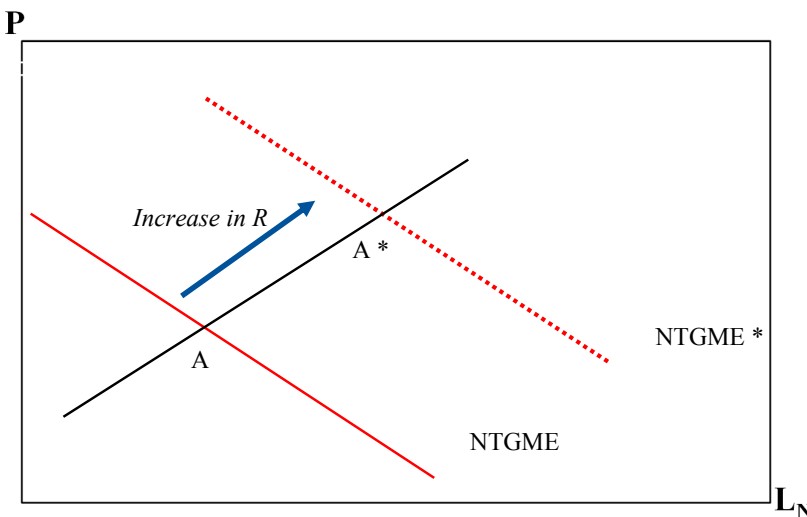

**Figure 2.** Dynamics of the nontradable sector in Dutch Disease Theory. Increasing *R* (rent revenue) will lead to an increase in the price of non-tradeable (*Pn*) and Labor Demand for non-tradeable (*Ln*).

There is a positive relationship between $P$ and the amount of rent revenue ($R$) spent in the domestic economy. This relationship can be represented as:

$$P = 1 + \theta(R), \text{ where} \frac{\partial(\theta(R))}{\partial R} > 0.$$

A8: There are $N_N$ agents focused on the production of nontradable goods. The ruler is also active in nontradable businesses with a share of $\alpha$ which is assumed to be fixed for both periods.

A9: The ruler is a significant economic actor in a rentier economy through controlling disbursement of rent revenues ($R$). As stated in 8th assumption, he owns a share in the nontradable sector along with private agents. As suggested in the literature, the loss of the ruler's economic power could result in loss of his political power, which is unacceptable in a rentier state, which described in Figure 3 [29]:

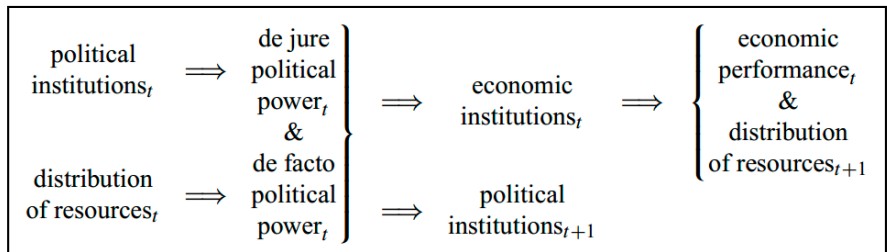

**Figure 3.** Economic and political interlink based on the SCT.

Therefore, the ruler will tolerate the economic diversification as long as his economic control (rents + ruler-owned nontradable) will not be less than a certain share of the total economy. It is stated that "*when a particular group is rich relative to others, this will increase its de facto political power and enable it to push for economic and political institutions favorable to its interests*" [8]. It is assumed that the ruler will not compromise from controlling a certain share ($\zeta$) of the economy in order to guarantee that there is no group that can challenge its status-quo defined, such as:

$$R + \alpha \times \left(P \times \sum X_N = P \times C_N\right) \geq \zeta \times Y, \text{ where } 0 \leq \zeta \leq 1.$$

The share of the ruler in nontradable sectors ($\alpha$) is a function of his desired controlling share in the economy ($\zeta$). Depending on the size of R relative to the whole economy ($Y$), the ruler controls a much larger stake ($\zeta$) in the economy. For example, this was the case for the early development phase of the 1970s and 80s in the GCC states [6,38]. Advancement in state security apparatuses and integration of the GCC economies with the rest of the world may have decreased the controlling share of the ruler ($\zeta$).

A10: Total number of agents is equal to $N = N_T + N_N$, which is a function of the total labor force ($L$) as $N = f(L)$ with $\frac{\partial(N=f(L))}{\partial L} > 0$. For a two-period economy, there will be no limitation to the number of agents in both sectors as long as the ruler allows.

A11: There are three sources of employment for labor: Government ($L_G$), tradable ($L_T$), and nontradable ($L_N$). There is a wage premium in government and tradable sectors compared to the wage in nontradable sectors. The premium for wages in public employment is part of a rentier agreement resulting from disbursing of revenue from natural resources. Premium public wages are a norm across the GCC states albeit varying degree among the members [39]. Since most of the workforce in nontradable sectors is from abroad and especially low-income countries, we assume an average wage, which is equal to 1 chosen as numeraire in A2. Following a similar study, a premium wage should be given in tradable sectors to attract skillful workers both from national and abroad [12]. Therefore, the wage of an employee in the public sector is $1 + u$ and wage in the tradable sector is $1 + v$ while it is 1 in nontradable sectors.

$$Total\ Labor\ Force\ (L) = L_G + L_T + L_N.$$

A12: Since there is very limited taxation in those rentier states of GCC as a part of rentier agreement, the financing of wages for public sector employees is performed by revenue from natural resources denoted as *R* [36]:

$$L_G \times (1 + u) = R.$$

The premium wage paid in the public sector ($W_G$) will be $u = \frac{R}{L_G} - 1$

Premium public wages in the GCC states are the norm, due to high resource revenues and the rentier agreement [40]. When there is a decrease in rent revenue (*R*), the ruler will decrease public employment or wages to balance the government budget both of which are undesirable for the stability of the current political system. Employment of citizens in premium-paid tradable sectors can alleviate this dilemma faced by the ruler. The ruler will aim economic diversification towards tradable sectors to ease pressure on the government budget as long as its share in the economy not compromised stated at the 9th assumption.

A13: Consumers have utility from consumption of both the tradable ($C_T$) and nontradable ($C_N$) goods as a Cobb-Douglass preference as we assume:

$$U = \left( C_T^{\frac{1+\varepsilon}{\varepsilon}} + C_N^{\frac{1+\varepsilon}{\varepsilon}} \right)^{\frac{\varepsilon}{1+\varepsilon}},$$

where $\varepsilon$ is to be the elasticity of substitution between tradable and nontradable goods.

A14: The number of agents is restricted in a rentier state as a part of "rentier agreement". Only the agents who have a license granted by the ruler can operate in nontradable sectors. Profit of a private agent will depend on the difference between its revenue and cost. For an agent in tradable sectors producing $X_T$ amount of good, his profit ($\Pi_T$) is the following based on the assumptions of A5 for revenue and A11 for the cost of labor:

$$\Pi_T = X_T \times 1 - \left( \frac{X_t + F}{\beta} \right) \times (1 + v).$$

When the total consumption of tradable goods and services is $C_T$, then this demand is evenly distributed for each tradable sector (*K*):

$$X_T = \frac{C_t}{K}.$$

The profit of an agent in nontradable sectors depends on the difference between its revenue and cost as well. Labor cost in nontradable sectors is equal to 1 following the assumption A11. The price of nontradable goods ($P_N$) is determined through Dutch Disease dynamics ($P = 1 + \theta$) based on assumption A7. The amount of production and profit of a nontradable agent can found as:

$$X_N = C_N/N_N, \ \Pi_N = \frac{C_N \times P - C_N}{N_N}.$$

*2.3. Decision Variables and Scenarios of the Model*

This model is a two-period event between the state and private agents. There is an exogenous shock at the end of the first period where rent revenue (*R*) decreases to half (*R*/2) for the second period. Private agents will choose either in nontradable sectors or tradable sectors depending on the profitability of these sectors with respect to each other. The ruler is assumed to have shared in nontradable sectors in addition to his control of rent revenues (*R*) to keep his rentier patronage and political authority safe. We assume that his share in nontradable sectors ($\alpha$) will be minimum but guarantee that his controlling share in the economy ($\zeta$) won't be less than a certain ratio as defined in assumption 9.

In response to decreasing in *R* for the second period, the private agents will decide whether to stay in nontradable sectors or switch to tradable sectors subject to the ruler's approval. The ruler will

decide following as the only economic policy authority (issuance of business license) at the end of the first period:

- Number of private agents operating in the nontradable sector for the second period ($N_{N2}$),
- Number of private agents operating in the tradable sector for the second period ($N_{T2}$).

The ruler's objective will be securing its minimum controlling share in the economy ($\zeta$) and supporting economic diversification into tradable sectors as long as its economic patronage not undermined. For the ruler, the economic diversification to tradable sectors is desirable to lessen the economy's dependence on revenue from natural resources and to create high-paying jobs for the citizens. However, economic diversification into tradable is secondary to the objective of holding its political power. Table 1 describes important decision variables, exogenous shocks, and parameters of the model.

**Table 1.** Decision variables, exogenous shock, and parameter of the model.

| The Ruler Decides Following to Maximize Economic Diversification at the End of $t = 1$ | Exogenous Shock | Parameters |
|---|---|---|
| $N_{N2}$ $N_{T2}$ | $R \rightarrow R/2$ $(P{:}1 + \theta \rightarrow 1 + \theta/2)$ | $F, \zeta$ |

The first scenario in this model for the second period is determined by profitability perspective for agents in tradable sectors versus to nontradable sectors. The profitability in these sectors depends on the parameters listed in Table 1. Firstly, there are two potential scenarios, due to the uncertainty of upfront investment (fixed cost) level of the tradable sector ($F$) as a parameter. In the first scenario, actual fixed cost ($F$) is higher than the threshold fixed cost ($F^*$) required for the private agents to make any profit. There will be no private agent in tradable sectors and with no diversification into tradable. The number of private agents in nontradable sectors is subject to the ruler's approval. The ruler will not block any number of private agents as long as his controlling share ($\zeta$) is not compromised.

In the second scenario, we assume that actual fixed cost of tradable sectors ($F$) is lower than the threshold level ($F^*$) and even a single agent will make a profit if it switches to that sector. However, the private agents in nontradable sectors from the first period wouldn't prefer to switch to tradable sectors, due to very high profits in nontradable sectors. This is due to the fact that actual fixed costs in tradable sectors ($F$) is bigger than the threshold cost level ($F^{**}$) that would equate the profits in both sectors. This would result in a similar outcome in the first scenario. There will be no economic diversification into tradable even this sector is profitable.

In the third scenario, the fixed cost of tradable sectors ($F$) is lower than the threshold level of fixed cost ($F^{**}$) such that a certain number of private agents consider switching to the tradable sectors, due to attractive profits. The second parameter for the rulers controlling share preference ($\zeta$) becomes an important factor in this scenario since the ruler may take an action to preserve its controlling share in the economy. The optimal number of private agents willing to switch to tradable sectors ($N_{T2}^*$) may be higher than the ruler can tolerate a maximum ($N_{T2,max}$). This is due to the fact that his preference of controlling share in the economy ($\zeta$) is higher than the threshold controlling level ($\zeta^*$) needed to accommodate an optimal number of agents in tradable sectors ($N_{T2}^*$). The economic diversification towards tradable sectors would be limited and below Pareto optimal.

In the fourth scenario, the number of private agents willing to switch ($N_{T2}^*$) may be less than the ruler can tolerate a maximum ($N_{T2,max}$). This is due to his preference of a relatively lower controlling share in the economy ($\zeta$) compared to the threshold controlling share ($\zeta^*$) need to accommodate an optimal number of agents in tradable sectors ($N_{T2}^*$). The economic diversification towards tradable sectors would not be limited and actively pushed further diversification towards tradable by the ruler. The reason the ruler fully supports this diversification is that this will ease the pressure on the public finances with some citizens employed in tradable sectors. The following decision graph in Figure 4

summarizes potential cause and result of each scenario that may emerge in a rentier economy under declining rent revenues.

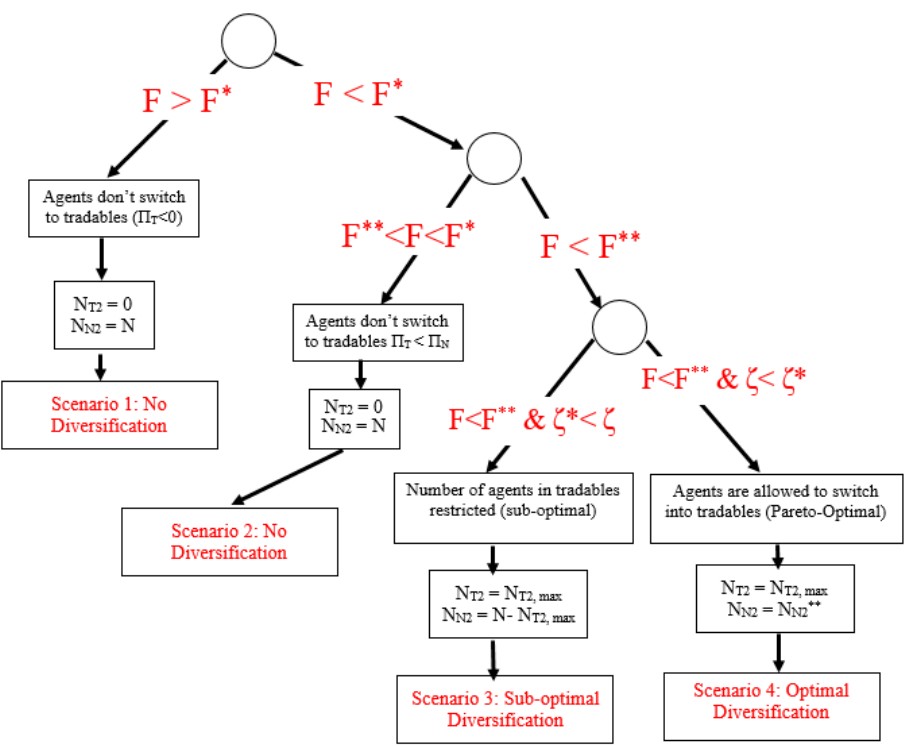

**Figure 4.** The model's proposed dynamics and potential scenarios.

## 3. A Solution of the Model and the Potential Scenario Outcomes for Economic Diversification

We start solving the model by finding the preferred consumption of the population from the tradable and nontradable sectors. We use a constant elasticity of substitution function with the Cobb-Douglas style in which it is impossible to substitute all tradable goods with nontradable good or vice versa. This makes sense since a person would need to consume a combination of those goods in his daily life. Depending on the relative prices of each good, there will be an increase in the consumption of relatively more inexpensive goods in the consumption bundle. Once the relative amount of goods to be consumed from both the tradable and nontradable sectors are determined with respect to each other, we calculate the demand to this consumption by all the national income, which include wages of the population working in public, nontradable, and tradable sectors, and the profits of private agents in both sectors. Although the private agents are on the production site, we also assume them to be effective consumers of the goods and services they and others produce.

In order to find the true size of total demand (national income), we should find a number of agents in tradable sectors where two scenarios and their potential scenarios come into the dynamics of the model. Before going into those scenarios, we solve a baseline model for the first period and lay out the size of the economy which is assumed to be equal to total demand or total consumption. Once the basic structures of the model in place, we will go the second period and solve the model for each potential scenario.

The demand side of the economy is shared between tradable and nontradable goods/services by the following utility (*U*) optimization problem subject to the budget constraints of consumers:

$$\text{Max } U = \left( C_T^{\frac{1+\varepsilon}{\varepsilon}} + C_N^{\frac{1+\varepsilon}{\varepsilon}} \right)^{\frac{\varepsilon}{1+\varepsilon}}.$$

Subject to

$$C_T + P \times C_N \leq L_G \times (1 + u) + L_T \times (1 + v) + L_N. \tag{1}$$

Solving for the Lagrangian ($L = C_T^{1/3} \times C_N^{2/3} + \lambda \times (L_G \times (1 + u) + L_T \times (1 + v) + L_N - C_T - P_N \times C_N)$) yields the optimal amounts of tradable ($C_T$) and nontradable ($C_N$) to be consumed with respect to each other as:

$$C_T = P^{-\varepsilon} \times C_N, \text{ or } C_N = P^{\varepsilon} \times C_T. \tag{2}$$

In order to find the size of the economy ($Y$), we need to equate domestic income to domestic demand. The model does not allow for savings and does not require investment for production with exception of fixed cost in tradables.

$$Y = \text{National Income (NI)} = \text{National Consumption (NC)} \tag{3}$$

Wage of all workers and profits of agents constitute the national income:

NI = Wage of Workers in the public sector (R) + Wage of Workers in the nontradable sectors + Wage of Workers in the tradable sectors + Value Added (profits of agents in tradable + nontradable sectors),

$$\text{NI}: Y = L_G \times (1 + u) + L_T \times (1 + v) + L_N + N_N \times \pi_N + N_T \times \pi_T + R. \tag{4}$$

It should be noticed that wage of labor and profit of private agents in each sector is equal to the value created in each sector respectively:

$$L_G \times (1 + u) = R, \quad L_N + N_N \times \pi_N = P \times C_N, \quad L_T \times (1 + v) + N_T \times \pi_T = \frac{N_T}{K} \times C_T,$$

which leads to NI to be rewritten as:

NI: $Y = P \times C_N + \frac{N_T}{K} \times C_T + R$, replacing $C_N$ with $P^{\varepsilon} \times C_T$ from Equation (2):

NI: $Y = P^{1+\varepsilon} \times C_T + \frac{N_T}{K} \times C_T + R$

National consumption (NC) is simply found by adding the total consumption of tradable and nontradable goods:

$$\text{NC}: Y = C_T + P \times C_N = C_T + P^{1+\varepsilon} \times C_T = C_T \times (1 + P^{1+\varepsilon}). \tag{5}$$

NI must be equal to NC for each period:

$$P^{1+\varepsilon} \times C_T + \frac{N_T}{K} \times C_T + R = C_T \times (1 + P^{1+\varepsilon}), \text{ where } C_T = \frac{K \times R}{K - N_T} \text{ and } C_N = P^{\varepsilon} \times \left( \frac{K \times R}{K - N_T} \right),$$

and the size of the economy ($Y$) becomes:

$$Y = C_T + C_T P^{1+\varepsilon} = (1 + P^{1+\varepsilon}) \times \left( \frac{K \times R}{K - N_T} \right). \tag{6}$$

It is very obvious that economy grows with increasing diversification or more local production of tradable goods and services ($\frac{\partial Y}{\partial N_T} > 0$) and higher wages to public employees following increase in revenue from natural resources ($R$).

Having established the basics of the model, we can start analyzing how a rentier economy works. There are $N$ agents divided into tradable ($N_P$) and nontradable ($N_N$) sectors. Profit of an agent in tradable ($\pi_T$) and nontradable sectors ($\pi_N$) is as follows respectively:

$$\pi_T = \frac{C_T}{K} - \left( \frac{C_T}{K \times \beta} + \frac{F}{\beta} \right) \times (1 + v) = \frac{R}{K - N_T} - \left( \frac{R}{(K - N_T) \times \beta} + \frac{F}{\beta} \right) \times (1 + v), \tag{7}$$

$$\pi_N = (1 - \alpha) \times \frac{C_N \times P - C_N}{N_N} = \frac{C_N \times (1 + \theta - 1)}{N_N} = (1 - \alpha) \times \frac{(1 + \theta)^{\varepsilon} \times (\theta) \times K \times R}{(K - N_T) \times N_N}. \tag{8}$$

Economy at first period:

We assume that the amount of rent revenue to be spent in the domestic economy ($R_1$) is $R$ and the scenario price level of nontradable goods and services is $P_1 = 1 + \theta$.

Following Equation (6), the size of the economy $Y_1$ will be:

$$Y_1 = C_{T1} + P^{1+\varepsilon} \times C_{T1} = \left(1 + (1 + \theta)^{1+\varepsilon}\right) \times \left(\frac{K \times R}{K - N_{T1}}\right), \tag{9}$$

where $C_{N1} = (1 + \theta)^\varepsilon \times \left(\frac{K \times R}{K - N_{T1}}\right)$; and $C_{T1} = \left(\frac{K \times R}{K - N_{T1}}\right)$

In assumption A9, we stated that the ruler must control at least half of the economy by controlling $R$ and owning a share in nontradable ($\alpha$) to retain his political power. His share ($\alpha$) is fixed for both the first and second periods which must satisfy:

$$R + \alpha \times ((1 + \theta) \times C_{N1}) \geq \zeta \times \left(Y_1 = \left(1 + (1 + \theta)^{1+\varepsilon}\right) \times \left(\frac{K \times R}{K - N_{T1}}\right)\right),$$

$R + \alpha \times ((1 + \theta) \times (1 + \theta)^\varepsilon \times \left(\frac{K \times R}{K - N_{T1}}\right)) \geq \zeta \times ((1 + (1 + \theta)^{1+\varepsilon}) \times \left(\frac{K \times R}{K - N_{T1}}\right))$ by cancelling R from each part:

$\frac{K - N_{T1}}{K} + \alpha \times (1 + \theta)^{1+\varepsilon} \geq \zeta + \zeta \times (1 + \theta)^{1+\varepsilon}$ which simplifies to the following:

$$\alpha \geq \left(\frac{N_{T1}}{K} - 1 + \zeta\right) \times \frac{1}{(1 + \theta)^{1+\varepsilon}} + \zeta. \tag{10}$$

Therefore, the minimum level of the ruler's share in nontradable sectors can be defined as:

$$\alpha_{\min} = \left(\frac{N_T}{K} - 1 + \zeta\right) \times \frac{1}{(1 + \theta)^{1+\varepsilon}} + \zeta. \tag{11}$$

**Remark 1.** *The increase in revenue from natural resources ($\Delta R$) also increases $\alpha_{min}$. We know that $\theta$ is a function of R from assumption A7:*

$$\frac{\partial \alpha_{min}}{\partial R} = \frac{\partial \alpha_{min}}{\partial \theta} \times \frac{\partial \theta}{\partial R} > 0, \text{ since } \frac{\partial \alpha_{min}}{\partial \theta} > 0 \text{ and } \frac{\partial \theta}{\partial R} > 0.$$

This indicates that increasing resource revenue ($R$) pushes the economy's balance towards nontradable sectors by increasing its prices ($P$) and the profitability of this sector. As a result, the ruler should increase its share in nontradable ($\alpha$) at the expense of private businessmen/agents to preserve its control of the economy and political spectrum. The ruler will allow citizens to pursue business activities in nontradable sectors further as a part of "rentier agreement" as long as his own share exceeds $\alpha_{min}$. The rulers of GCC states own companies or shares in various nontradable firms in the economy in all these countries [12]. They also support select citizens and merchant families into remaining nontradable sectors by giving them exclusive licenses and various forms of credit support [40,41]. The famous "Kafala" (sponsorship) where a foreign agent should have a local partner to do business is an epitome of this policy [29].

**Remark 2.** *The change in the share of agents in tradable sectors ($\Delta N_T$) increases $\alpha_{min}$. This is very straightforward as:*

$$\frac{\partial \alpha_{min}}{\partial N_T} = \left(\frac{1}{K}\right) \times \frac{1}{(1 + \theta)^{1+\varepsilon}} > 0.$$

Since both terms in multiplication are positive.

This conclusion is intuitive as it has been shown that economic power is a prerequisite for holding political power [39]. In return, an absolute authority will not tolerate private agents to grow beyond a threshold point where they control the majority of the economy known as the SCT theory.

**Remark 3.** *The effect of a change in share of agents in tradable ($\Delta N_T$) on $\alpha_{min}$ decreases in magnitude with higher revenues from natural resources (R). This occurs through R's effect on the price of nontradable ($\theta$). Based on results from Remark 2:*

$$\frac{\frac{\partial \alpha_{min}}{\partial N_T}}{\partial R} = \frac{\frac{\partial \alpha_{min}}{\partial N_T}}{\partial \theta} \times \frac{\partial \theta}{\partial R} = \frac{\partial \left( \left(\frac{1}{K}\right) \times \frac{1}{(1+\theta)^{1+\varepsilon}} \right)}{\partial \theta} = \left\{ \underbrace{\left(\frac{1}{K}\right) \times (-1-\varepsilon) \frac{1}{(1+\theta)^{2+\varepsilon}}}_{-} \times \underbrace{\frac{\partial \theta}{\partial R}}_{+} \right\} < 0.$$

This indicates that the higher the revenues from natural resources ($R$), the less the ruler will be concerned about the change in a number of agents in tradable sectors ($\Delta N_T$).

Similar to the first period, the ruler's share must satisfy the following condition:

$$\alpha \geq \left( \frac{N_{T2}}{K} - 1 + \zeta \right) \times \frac{1}{(1 + \theta/2)^{1+\varepsilon}} + \zeta. \tag{12}$$

**Remark 4.** *The higher the ruler prefers a controlling stake in the economy ($\zeta$), the higher his minimum acceptable share ($\alpha_{min}$) will be in the nontradable sectors. This is straightforward inference since:*

$$\frac{\partial \alpha_{min}}{\partial \zeta} = \frac{1}{(1 + \theta)^{1+\varepsilon}} + 1 > 0, \text{ Since } \frac{1}{(1 + \theta)^{1+\varepsilon}} > 0.$$

*3.1. Potential Scenarios for Economic Diversification in a Rentier State*

An exogenous shock happens at the end of the first period where rent revenue from natural resources ($R$) decreases to half ($R/2$) for the second period. Accordingly, the price of nontradable ($P_2$) adjusts to this reality by decreasing to ($1 + \theta/2$) from ($1 + \theta$). After these developments, the ruler will decide for the second period the following:

- Number of private agents operating in the nontradable sector for the second period ($N_{N2}$),
- Number of private agents operating in the tradable sector for the second period ($N_{T2}$).

In response to the ruler's actions, eligible private agents will decide whether:

- To stay in nontradable sectors,
- To move into tradable sectors in order to maximize their expected profits ($\pi_{NT2}$ or $\pi_{T2}$).

As highlighted in Section 3.2, there two possible scenarios with four potential scenarios in total for the second period. We will study each one of those scenarios in order to infer what can be concluded for a rentier state once the rent revenues start falling.

**Scenario 1.** *If the fixed cost of private agents in tradable sectors (F) is bigger than the threshold fixed cost (F\*) in period 2 as a result of declined rent revenues, then there may be no single agent ($N_{T2} = 0$) willing to make an investment into tradable productions. In this case, an agent willing to make an investment into tradable sectors will realize that his investment will have a negative profit, i.e., $\pi_{NT2=1} < 0$. We define the threshold fixed cost level as the cost level where a single agent in nontradable sectors cannot make a profit:*

$$\Pi_{Nt2=1} = \frac{R/2}{K-1} - \left( \frac{R/2}{(K-1) \times \beta} + \frac{F^*}{\beta} \right) \times (1 + v) = 0.$$
$$F* = \frac{(\beta - 1 - v) \times R/2}{(K-1) \times (1+v)}$$

For that scenario to emerge, the following inequality must hold for the fixed cost of *F*:

$$F \; > \; F* \; = \; \frac{(\beta - 1 - v) \times R/2}{(K-1) \times (1+v)}.$$
(13)

Whenever *F* is bigger than *F\**, then there will be no investment into tradable sectors by the private agents. This may occur when rent revenues (*R/2*) are low (small economy), productivity differential from premium wage ($\beta - 1 - v$) is not very high to overcome high fixed cost (*F\**). The economy may get stuck in complete dependence on imports for tradable goods and services with no diversification at all. Since value addition occurs in tradable (apart from natural resources), lack of domestic tradable sectors is described as "poverty trap" [42,43]. This is represented as point A in Figure 5:

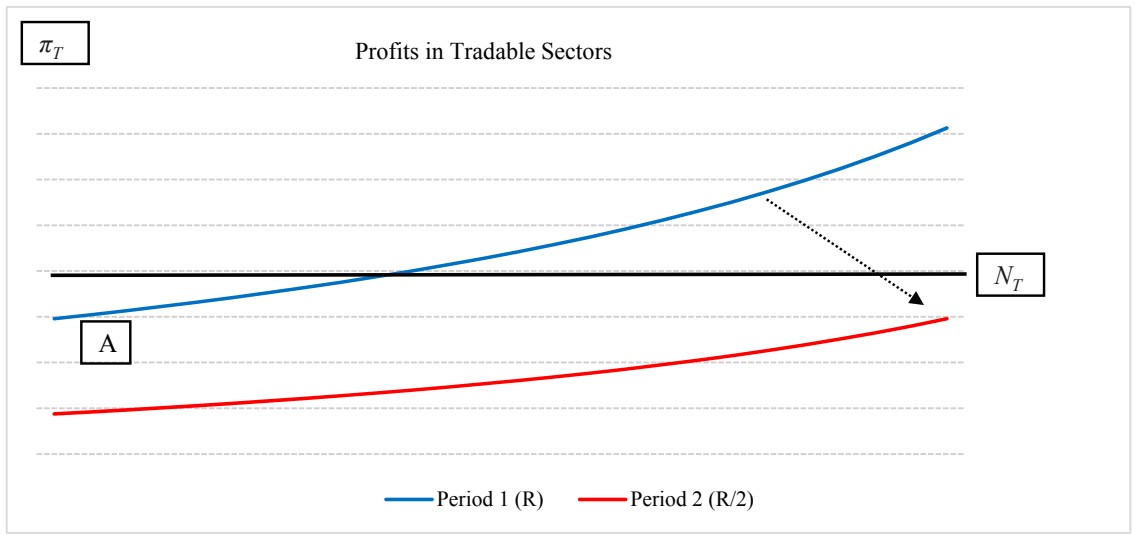

**Figure 5.** Profit of a tradable agent vs. number of tradable agents in 1st and 2nd periods.

A rentier economy at point A will totally depend on revenue from natural resources as the main value addition in the economy. In this first scenario, $N_{T2} = 0$ and $N_{N2} = N_2$ and where the size of the economy is as follows:

$$Y_2 \; = \; \left(1 + (1 + \theta/2)^{1+\varepsilon}\right) \left(\frac{K}{K - N_{T2} = 0}\right) \times \frac{R}{2} \; = \; \left(1 + (1 + \theta/2)^{1+\varepsilon}\right) \times \frac{R}{2}.$$

Based on Equation (12), the ruler's share in nontradable sectors (constant for both periods) will satisfy the following condition for holding political power:

$$\frac{R}{2} + \alpha \times \left(C_N \times \left(1 + \frac{\theta}{2}\right)\right) \geq \zeta \times Y_2,$$

$$\alpha_{min} = \left\{\left(\frac{N_{T1}=0}{K} - 1 + \zeta\right) \times \frac{1}{(1+\theta)^{1+\varepsilon}} + \zeta\right\} \geq \left\{\left(\frac{N_{T2}=0}{K} - 1 + \zeta\right) \times \frac{1}{\left(1+\frac{\theta}{2}\right)^{1+\varepsilon}} + \zeta = (-1 + \zeta) \times \frac{1}{\left(1+\frac{\theta}{2}\right)^{1+\varepsilon}} + \zeta\right\}.$$
(14)

We know that $\frac{1}{(1+\theta)^{1+\varepsilon}} \leq \frac{1}{\left(1+\frac{\theta}{2}\right)^{1+\varepsilon}}$ and $-1 + \zeta \leq 0$ since $\zeta \leq 1$ which together prove that the inequality (14) holds.

The ruler does not need to impose any restriction on a number of private agents (entrepreneurs) in tradable sectors for the second period to preserve its political power. Figure 5 represents the shift in profitability of tradable sectors for the second period which indicates that "poverty trap" even harder to escape in the realm of declining revenue from natural resources. Table 2 summarizes the potential outcomes of scenario 1.

**Table 2.** The potential outcomes of the first scenario.

| Ruler's Decision at the End of $t = 1$ | Size of the Economy | Public Employment and Wages |
|---|---|---|
| $N_{N2}$: No action needed $N_{T2}$: No action needed | $Y_2 = (1 + (1 + \theta/2)^{1+\varepsilon}) \times R/2$ | $L_{G1} = L_{G2}$ $W_1 = 1 + u$, $W_2 = (1 + u)/2$ |

**Scenario 2.** *If the fixed cost of private agents in tradable sectors (F) is smaller than the threshold fixes cost (F\*) in period 2, then there may be some agents willing to make an investment into tradable productions. As we have shown in the scenario above that threshold fixed cost level (F\*) is:*

$$F* = \frac{(\beta - 1 - v) \times R/2}{(K - 1) \times (1 + v)}.$$

For tradable agents to make a profit, the following inequality must hold for the fixed cost of *F*:

$$F < F^* = \frac{(\beta - 1 - v) \times R/2}{(K - 1) \times (1 + v)}. \tag{15}$$

Therefore, even a single agent will make a profit by entering into tradable sectors as long as inequality (15) holds. However, the following inequality may occur in profits of tradable vs. nontradable sectors which will disincentivize the private agents:

$$\Pi_{T2=1} < \Pi_{N2=N-1}. \tag{16}$$

Inequality of (16) states that even when all but a single agent is in nontradable sectors ($N_{N2} = N - 1$), each of their profit is bigger than the single agent in tradable sectors, due to still high fixed costs (*F\*\**). The following equation yields the threshold fixed cost (*F\*\**) that will induce some of the agents to switch into tradable sectors from nontradable sectors:

$$\frac{R/2}{K} - \left(\frac{R/2}{(K) \times \beta} + \frac{F^{**}}{\beta}\right) \times (1 + v) = (1 - \alpha) \times \frac{(1 + \theta/2)^\varepsilon \times (\theta/2) \times K \times R/2}{(K) \times N}. \tag{17}$$

After rearranging the terms:

$$F** = R \times \frac{N \times (\beta - 1 - v) - K \times \beta \times (1 - \alpha) \times (1 + \theta/2)^\varepsilon \times (\theta/2)}{2 \times K \times N \times (1 + v)}.$$

When fixed cost is small enough to yield a profit (*F < F\**) but big enough to disincentivize the agents from switching nontradable sectors to tradable sectors (*F\*\* < F < F\**):

$$F** = R \times \frac{N \times (\beta - 1 - v) - K \times \beta \times (1 - \alpha) \times (1 + \theta/2)^\varepsilon \times (\theta/2)}{2 \times K \times N \times (1 + v)} < F < F^* = \frac{(\beta - 1 - v) \times R/2}{(K - 1) \times (1 + v)}.$$

There will be no diversification similar to scenario 1. This scenario would be likely to hold when the productivity of labor in tradable sectors is small compared to the wages ($\beta - 1 - v$). Figure 6 represents the dynamics of first scenarios where all the agents (*N*) choose nontradable sectors, due to higher profits.

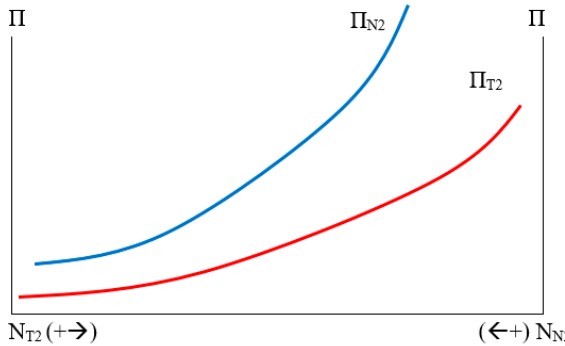

**Figure 6.** Profits of Agents in tradable and nontradable sectors in period 2.

Before explaining the dynamics of this scenario, it is helpful to describe why the profit curves for the private agents ($\Pi_{N2}$-blue nontradable, $\Pi_{T2}$-red for tradable behave, as shown in Figure 6). We have shown in Equations (7) and (8) the profits of an agent in tradable sectors and in nontradable sectors respectively. For tradable sectors, it is clear that profits will increase with more agents in this sector, due to increasing returns to scale production nature since

$$\frac{\partial \pi_T}{\partial N_T} = \frac{R}{(K - N_T)^2} - \left( \frac{R}{(K - N_T)^2 \times \beta} \right) \times (1 + v) > 0,$$

since $\beta > 1 + v$ by assumption and $\frac{R}{(K-N_T)^2} \left( 1 - \frac{1+v}{\beta} \right) > 0$, accordingly.

Therefore, the profit of an agent in tradable sectors increases through the right axis due to an increase in the number of total agents in this sector. For nontradable sectors, profits decrease with an increasing number of total agents in this sector since it does not add value and shows constant returns to scale production:

$$\frac{\partial \pi_N}{\partial N_N} = -(1 - \alpha) \times \frac{C_N \times P - C_N}{N_N^2} < 0.$$

The more non-tradable agents in the economy, the less profit per private agent in the nontradable sectors. Hence, the profits of agents in nontradable sectors decrease through the left axis with an increasing number of agents in this sector.

In this second scenario, there will be no agent in the tradable sector similar to the first scenario, due to lower profits than nontradable, which would result in no diversification. Based on Equation (11), a minimum share of the ruler ($\alpha$) in nontradable sectors to preserve political power would be:

$$\alpha_{\min} = \left( \frac{N_{T2} = 0}{K} - 1 + \zeta \right) \times \frac{1}{(1 + \theta)^{1+\varepsilon}} + \zeta = (\zeta - 1) \times \frac{1}{(1 + \theta)^{1+\varepsilon}} + \zeta. \tag{18}$$

Due to exogenous shock to revenue from natural resources ($R \to R/2$), profits of agents in nontradable and tradable sectors will decrease even further. Profits of nontradable agents ($\Pi_{N2}$) will decrease more compared to the profits of tradable agents ($\Pi_{T2}$) following the inequality of (17):

$$\frac{\partial \Pi_T}{\partial R} = \frac{1/2}{K - N_T} - \left( \frac{1/2}{(K - N_T) \times \beta} \right) \times (1 + v) < \frac{\partial \Pi_N}{\partial R} = (1 - \alpha) \times \frac{(1 + \theta/2)^{\varepsilon} \times (\theta/2) \times K \times 1/2}{(K - N_T) \times N_N}.$$

Therefore, $\Delta R = -R/2$ case will have a more negative effect on the profits of a nontradable agent than profits of a tradable agent. This result is also intuitive since a decrease in $R$ will not just decrease the consumption of nontradable goods ($C_N$) but also its price ($P$). In fact, this could be a reasonable driver for nontradable agents to invest in the tradable sector gradually, as can be expected naturally.

The ruler's share in nontradable sectors will not be overrun in the second period since its minimum share in period 1 ($\alpha_{min,1}$) will be larger than minimum share needed in the second run ($\alpha_{min,2}$):

$$\alpha_{min} = \left\{ \left( \frac{N_{T1}=0}{K} - 1 + \zeta \right) \times \frac{1}{(1+\theta)^{1+\varepsilon}} + \zeta \right\} \geq \left\{ \left( \frac{N_{T2}=0}{K} - 1 + \zeta \right) \times \frac{1}{\left(1+\frac{\theta}{2}\right)^{1+\varepsilon}} + \zeta = (-1+\zeta) \times \frac{1}{\left(1+\frac{\theta}{2}\right)^{1+\varepsilon}} + \zeta \right\},$$

Same as inequality (14).

Therefore, the ruler does not need to take any decision regarding the number of agents in nontradable ($N_{N2}$) and tradable ($N_{T2}$) sectors which summarized in Table 3.

**Table 3.** The outcomes of the second Scenario.

| Ruler's Decision at the End of $t = 1$ | Size of the Economy | Public Employment and Wages |
|---|---|---|
| $N_{N2}$: No action needed<br>$N_{T2}$: No action needed | $Y_2 = (1 + (1 + \theta/2)^{1+\varepsilon}) \times R/2$ | $L_{G1} = L_{G2}$<br>$W_1 = 1 + u,\ W_2 = (1 + u)/2$ |

**Scenario 3.** *In the third scenario, the following condition holds in the second period following the decrease in rent revenues ($R \to R/2$):*

$$\Pi_{T2} = \Pi_{N2} \tag{19}$$

Which translates into:

$$\frac{\frac{R}{2}}{K - N_{T2}^*} - \left( \frac{\frac{R}{2}}{(K - N_{T2}^*) \times \beta} + \frac{F_1}{\beta} \right) \times (1 + v) = (1 - \alpha) \times \frac{\left(1 + \frac{\theta}{2}\right)^{\varepsilon} \times \left(\frac{\theta}{2}\right) \times K \times \frac{R}{2}}{(K - N_{T2}^*) \times (N - N_{T2}^*)}$$

The solution of this equality yields:

$$N_{T2}^* = N - (1 - \alpha) \times \frac{\left(1 + \frac{\theta}{2}\right)^{\varepsilon} \times \left(\frac{\theta}{2}\right) \times K}{(\beta - 1 - v)} \tag{20}$$

If (19) holds, then $N_{T2} = N_{T2}^*$ and $N_{N2} = N - N_{T2}^*$ whereas $N_{T2}^*$ is the solution of the equality above. The potential size of the economy then becomes:

$$Y_2 = \left(1 + (1 + \theta/2)^{1+\varepsilon}\right) \left(\frac{K}{K - N_{T2}^*}\right) \times \frac{R}{2} \tag{21}$$

The equality condition above (20) states that it is attractive for a certain number of agents ($N_T \geq 1$) to invest in the tradable sectors rather than into the nontradable sectors. Figure 7 shows the graphical representation of the second potential scenarios where there are agents in both the tradable and nontradable sectors with their unit profits are equal.

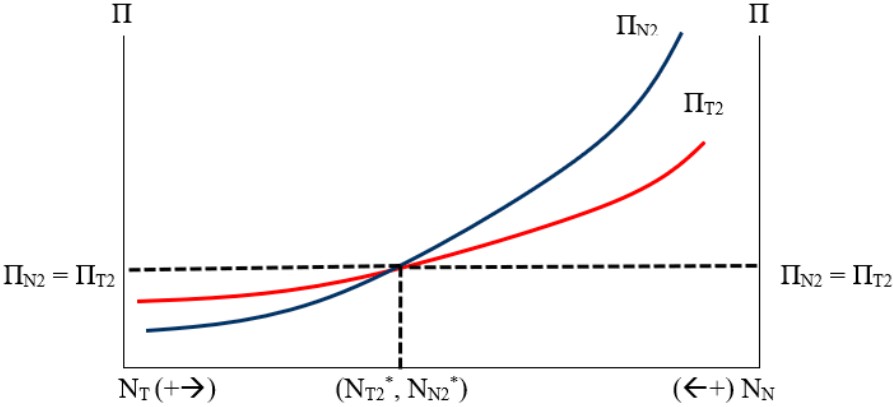

**Figure 7.** Profits of agents in the tradable and nontradable sectors in the third scenario.

In this scenario, a certain number of agents will switch to tradable from nontradable if the ruler grants them a license at the end of the first period. The ruler will not oppose this transition as long as his share in the economy is not compromised. Let say the ruler's share in the first period ($\alpha$) is such that it satisfies the minimum controlling share of the total economy preferred by the ruler ($\zeta$):

$$\alpha_{min} = \left( \frac{N_{T1} = 0}{K} - 1 + \zeta \right) \times \frac{1}{(1 + \theta)^{1+\varepsilon}} + \zeta = (\zeta - 1) \times \frac{1}{(1 + \theta)^{1+\varepsilon}} + \zeta.$$

In the second period, the number of tradable agents ($N_{T2}$) will be bounded by the following inequality:

$$\alpha_{min} = \left( \frac{N_{T1}=0}{K} - 1 + \zeta \right) \times \frac{1}{(1+\theta)^{1+\varepsilon}} + \zeta = (\zeta - 1) \times \frac{1}{(1+\theta)^{1+\varepsilon}} + \zeta \geq \left( \frac{N_{T2}}{K} - 1 + \zeta \right) \times \frac{1}{\left(1+\frac{\theta}{2}\right)^{1+\varepsilon}} + \zeta. \quad (22)$$

After rearranging the terms of inequality (22):

$$N_{T2} \leq K \times (1 - \zeta) \times \left( 1 - \left( \frac{1 + \frac{\theta}{2}}{1 + \theta} \right)^{1+\varepsilon} \right). \quad (23)$$

Since $K$ is a parameter, the allowable number of tradable agents in period 2 ($N_{T2}$) is positively related to the price premium of nontradable goods ($\theta$) which is a proxy for revenue from rent ($R$). If we define the maximum number of tradable agents to be allowed in period 2, such as:

$$N_{T2,\,max} = K \times (1 - \zeta) \times \left( 1 - \left( \frac{1 + \frac{\theta}{2}}{1 + \theta} \right)^{1+\varepsilon} \right), \quad (24)$$

$$\frac{\partial N_{T2,\,max}}{\partial R} \frac{\partial \theta}{\partial R} = \frac{\partial N_{T2,\,max} = K \times (1-\zeta) \times \left( 1 - \left( \frac{1+\frac{\theta}{2}}{1+\theta} \right)^{1+\varepsilon} \right)}{\partial \theta}$$

$$= K \times (1 - \zeta) \times (1 + \varepsilon) \times \underbrace{\left( \frac{\left(1 + \frac{\theta}{2}\right)^{1+\varepsilon}}{(1 + \theta)^{2+\varepsilon}} - \frac{1}{2} \times \frac{\left(1 + \frac{\theta}{2}\right)^{\varepsilon}}{(1 + \theta)^{1+\varepsilon}} \right)}_{+} \times \underbrace{\frac{\partial \theta}{\partial R}}_{+} > 0.$$

More importantly, the number of maximum tradable agents that the ruler can tolerate is negatively related to the desired controlling share of the economy ($\zeta$) since:

$$\frac{\partial N_{T2,max}}{\partial \zeta} = \frac{\partial N_{T2,max} = K \times (1 - \zeta) \times \left( 1 - \left( \frac{1+\frac{\theta}{2}}{1+\theta} \right)^{1+\varepsilon} \right)}{\partial \zeta} = -K \times \left( 1 - \left( \frac{1 + \frac{\theta}{2}}{1 + \theta} \right)^{1+\varepsilon} \right) < 0.$$

Therefore, the maximum tolerable level of tradable agents ($N_{T2,max}$) by the ruler has a direct negative relationship with his desired controlling share of the economy ($\zeta$). There emerge two potential scenarios in this case, which depends on the controlling share of the economy ($\zeta$) of the ruler. The ruler will block a certain number of agents switching into tradable sectors if scenario number of tradable agents in Figures 3–7 ($N_{T2}{}^*$) exceeds the maximum tolerable number of tradable agents ($N_{T2,max}$) given in Equation (24). This is labeled as the third scenario (third scenario).

If the maximum tolerable number of agents in tradable sectors ($N_{T2,max}$) is smaller than scenario number of agents ($N_{T2}{}^*$) found in Equation (20), due to preferred controlling share of the ruler ($\zeta$), then the ruler will intervene and restrict the number of agents in tradable sectors. The following equation yields the threshold share of the ruler ($\zeta^*$) where the maximum number of agents that he can allow in

tradable sectors ($N_{T2,max}$) is equal to the optimum number of agents that should switch to tradable sectors ($N_{T2}{}^*$):

$$N_{T2,\,max} = K \times (1 - \zeta) \times \left(1 - \left(\frac{1 + \frac{\theta}{2}}{1 + \theta}\right)^{1+\varepsilon}\right) < NT_2{}^* = N - (1 - \alpha) \times \frac{\left(1 + \frac{\theta}{2}\right)^{\varepsilon} \times \left(\frac{\theta}{2}\right) \times K}{(\beta - 1 - v)},$$

which yields the following result:

$$\zeta* = 1 - \frac{N - (1 - \alpha) \times \frac{\left(1+\frac{\theta}{2}\right)^{\varepsilon} \times \left(\frac{\theta}{2}\right) \times K}{(\beta - 1 - v)}}{K \times \left(1 - \left(\frac{1+\frac{\theta}{2}}{1+\theta}\right)^{1+\varepsilon}\right)}. \tag{25}$$

When the ruler's controlling share ($\zeta$) is higher than the threshold level ($\zeta^*$), then the economy will be diversified towards tradable sectors below its true potential and there will be no Pareto optimality:

$$\zeta* = 1 - \frac{N - (1 - \alpha) \times \frac{\left(1+\frac{\theta}{2}\right)^{\varepsilon} \times \left(\frac{\theta}{2}\right) \times K}{(\beta - 1 - v)}}{K \times \left(1 - \left(\frac{1+\frac{\theta}{2}}{1+\theta}\right)^{1+\varepsilon}\right)} < \zeta.$$

This result ($\zeta^* < \zeta$) can happen in three different ways in reference to the Equation (25). The high differential between productivity and wages ($\beta - 1 - v$) will yield higher profits to the tradable agents resulting in a bigger share for them in the economy. This will decrease the economic power of the ruler. That is one of the reasons why the ruler would exert entry barriers. From the Equation (25), it is obvious that threshold level since ($\zeta^*$) will decrease with increasing productivity-wage differential in tradable sectors since ($\frac{\partial \zeta^*}{\partial ((\beta - 1 - v))} < 0$). Therefore, the lower the threshold level of control needed to reach optimality ($\zeta^*$), the higher the chances that the ruler's preferred share ($\zeta$) will be higher than the threshold level.

Second, lower rent revenues result in lower prices for nontradable goods ($1 + \theta$) would also decrease the threshold level of economic control by the ruler ($\zeta^*$) since ($\frac{\partial \zeta^*}{\partial \theta} \frac{\partial \theta}{\partial R} > 0$). This means that declining rent revenues may force the ruler to exert further entry barriers especially in relatively less wealthy states in terms of rent revenues.

Finally, the share of the ruler in nontradable sectors ($\alpha$) increases the threshold level of economic control ($\zeta^*$). This means that the lower the ruler's share in the nontradable sectors, the more likely that he will exert entry barriers to the tradable sectors in the face declining rent revenues. This result is quite intuitive in the sense that the ruler's decreasing share in nontradable sectors may increase his actual preferred controlling share ($\zeta$) in the economy. This would result in more tolerance by him towards tradable sectors. Figure 8 represents the profitability dynamics of agents in both sectors for the third scenario:

In this scenario, putting a restriction on the number of agents in tradable sectors will distort the profits made by agents in nontradable vs. tradable sectors. Figure 8 shows that the profit of a single agent in the tradable sectors will be more than the profit of a single agent in nontradable ($\Pi_{T2} > \Pi_{N2}$). Restriction of the number of agents in the tradable section by the ruler will result in a smaller economy, due to less diversification into tradable in the second period. Decisions of the ruler and the level of economy for the third scenario is summarized in Table 4.

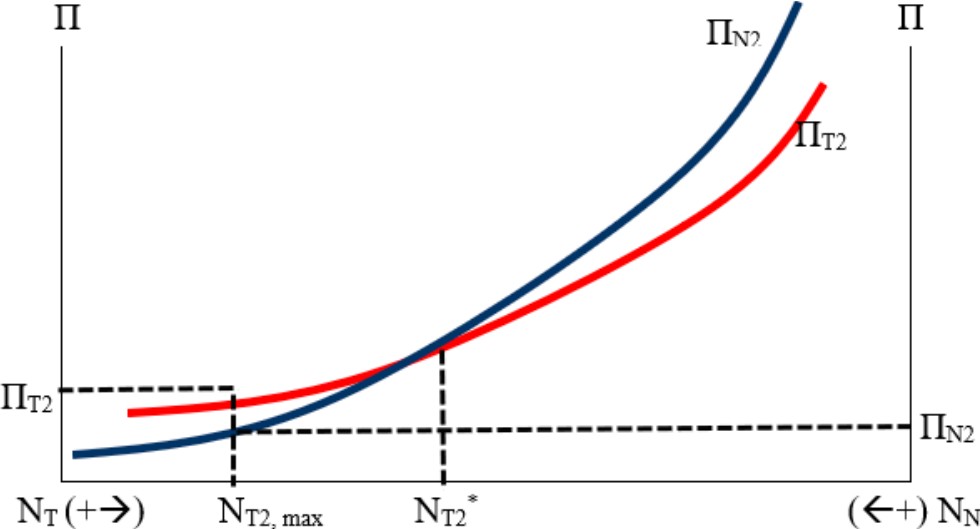

**Figure 8.** Profits of agents in the tradable and nontradable sectors in period 2 Scenario 3.

**Table 4.** The Outcomes of the third scenario (Scenario 3).

| Ruler's Decision At the End of $t = 1$ | Size of the Economy | Public Employment and Wages |
|---|---|---|
| $N_{N2}$: No action needed | $Y_2 = (1 + (1 + \theta/2)^{1+\varepsilon}) \left( \frac{K}{K - N_{T, \, max}} \right)$ | |
| $N_{T2}$: Restricted to $N_{T2,max}$ | $\times R/2 < (1 + (1 + \theta/2)^{1+\varepsilon})$ $\left( \frac{K}{K - N_{T2}^*} \right) \times R/2$ | $L_{G1} \geq L_{G2}$ $W_1 = 1 + u, (1 + u)/2 \leq W_2 \leq 1 + u$ |

If the maximum tolerable number of agents in tradable sectors ($N_{T2,max}$) is larger than scenario number of agents ($N_{T2}$*) found in Equation (19), then the ruler will not intervene and restrict the number of agents in tradable sectors. This will happen when:

$$N_{T2, \, max} = K \times (1 - \zeta) \times \left( 1 - \left( \frac{1 + \frac{\theta}{2}}{1 + \theta} \right)^{1+\varepsilon} \right) > NT_2* = N - (1 - \alpha) \times \frac{\left( 1 + \frac{\theta}{2} \right)^{\varepsilon} \times \left( \frac{\theta}{2} \right) \times K}{(\beta - 1 - v)}.$$

For that inequality to hold, the ruler's controlling share ($\zeta$) should be high enough such that:

$$\zeta < \zeta* = 1 - \frac{N - (1 - \alpha) \times \frac{\left( 1 + \frac{\theta}{2} \right)^{\varepsilon} \times \left( \frac{\theta}{2} \right) \times K}{(\beta - 1 - v)}}{K \times \left( 1 - \left( \frac{1 + \frac{\theta}{2}}{1 + \theta} \right)^{1+\varepsilon} \right)}. \tag{26}$$

The ruler's controlling share ($\zeta$) is lower than the threshold level required for optimal diversification ($\zeta$*). In this scenario, the economy will highly be diversified towards tradable sectors. This inequality will hold when the share of the ruler in nontradable ($\alpha$) is high, or the increasing returns to scale in tradable sectors compared to the wage premium ($\beta - 1 - v$) is not very high. In this third potential scenario, the ruler will not exert any entry barriers for the agents switching into tradable sectors. Figure 9 represents this case.

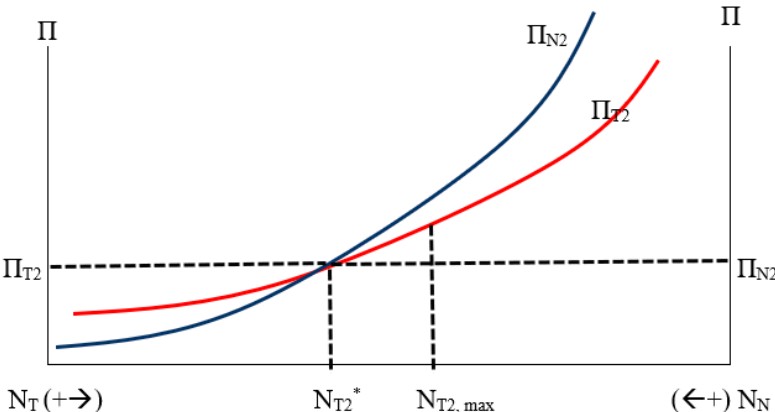

**Figure 9.** Agents in the tradable and nontradable sectors in period 2 Scenario 4.

In the fourth scenario, there will be fewer agents willing to invest in tradable sectors than the maximum the ruler can tolerate ($N_{T2}^* < N_{T2,\,max}$), due to the profitability dynamics of both sectors. In this case, the economy is still not at its maximum potential, due to the fact that:

$$Y_2 = \left(1 + (1 + \theta/2)^{1+\varepsilon}\right) \left(\frac{K}{K-N_{T2}^*}\right) \times R/2 < Y_2 = \left(1 + (1 + \theta/2)^{1+\varepsilon}\right) \left(\frac{K}{K-N_{T,\,max}}\right) \times R/2.$$

In order to entice more agents into tradable sectors and reach $N_{T2,max}$ meaning a higher diversification into tradable sectors, the ruler can increase the number of agents for nontradable sectors in the second period ($N_{N2}$). This decision will shift the profit curve of agents in nontradable sectors downwards, due to a higher number of agents in that sector (from $\Pi_{N2}$ to $\Pi_{N2}^*$), as shown in Figure 10.

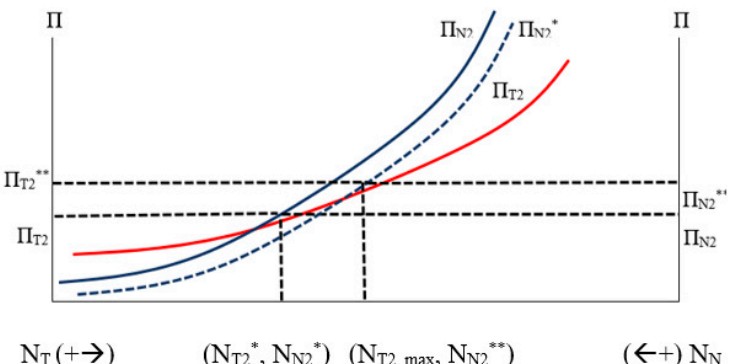

**Figure 10.** Profits of agents in the tradable and nontradable sectors in period 2 for Scenario 4.

If the ruler decides to allow for a large number of agents in the economy, some of those agents will shift to tradable sectors, due to the perception of higher profits there. The number of agents in tradable sectors will increase from $N_{T2}^*$ to $N_{T2,max}$, and their profits as well (from $\Pi_{T2}$ to $\Pi_{T2}^{**}$). Interestingly, the increase in the number of agents for nontradable sectors will increase profits of agents in the nontradable sectors since at scenario $\Pi_{N2}^{**} = \Pi_{T2}^{**} > \Pi_{N2} = \Pi_{T2}$. This is due to the fact that increase in the number of agents in tradable sectors (from $N_{T2}^*$ to $N_{T2,max}$) will expand the overall economy, due to increasing returns to scale nature of this sector. The ruler will benefit from the new scenario since some of the public employees will be transferred to high paying jobs in tradable sectors while its primary share in the economy will not be compromised.

This result ($\zeta < \zeta^*$) can happen in three different ways. The low differential between productivity and wages ($\beta - 1 - v$) will yield lower profits to the tradable agents resulting in a smaller share for them in the economy. This will increase the economic power of the ruler, which is one of the reasons

why he would exert entry barriers. From the Equation (25), it is obvious that threshold level ($\zeta^*$) will decrease with increasing productivity-wage differential in tradable sectors since ($\frac{\partial \zeta^*}{\partial((\beta-1-v))} < 0$). Therefore, a higher threshold level of controlling ($\zeta^*$) needed to reach optimality, the lower the chances that the ruler's preferred share ($\zeta$) will be higher than the threshold level. This would lead to higher chances of optimal diversification in the economy towards tradables.

Secondly, the share of the ruler in nontradable sectors ($\alpha$) increases the threshold level of economic control ($\zeta^*$). This means that the higher the ruler's share in the nontradable sectors, the less likely that he will exert entry barriers to the tradable sectors in the face declining rent revenues. This result is quite intuitive in the sense that the ruler's increasing share in nontradable sectors may lower his actual preferred controlling share ($\zeta$). This would result in more tolerance by him towards tradable sectors. Table 5 summarizes the outcome of the fourth scenario. Table 6 summarizes the results of all scenarios depending on the effects of selected parameters (*F* or $\zeta$).

**Table 5.** The Outcomes of the fourth scenario (scenario 4).

| Ruler's Decision At the End of $t = 1$ | Size of the Economy | Public Employment and Wages |
|---|---|---|
| $N_{N2}$: Increased to $N_{N2}^{**}$ from $N_{N2}^{*}$　　$N_{T2}$: Increased to $N_{T2,max}$ | $Y_2 = (1 + (1 + \theta/2)^{1+\varepsilon})$ $\left(\frac{K}{K-N_{T,\,max}}\right) \times R/2 > (1+ (1 + \theta/2)^{1+\varepsilon}) \left(\frac{K}{K-N_{T2}^*}\right) \times R/2$ | $L_{G1} \geq L_{G2}$ $W_1 = 1 + u,\ (1 + u)/2 \leq W_2 \leq 1 + u$ |

**Table 6.** Potential scenarios of the model.

| Scenarios | Binding Equations of the Scenario | Key Driver of the Scenario | Level of Diversification |
|---|---|---|---|
| Scenario 1 | $F < F^* = \frac{(\beta-1-v) \times R/2}{(K-1) \times (1+v)}$ | $F^* < F \rightarrow \Pi < 0$ | No diversification |
| Scenario 2 | $F^{**} < F < F^* = \frac{(\beta-1-v) \times R/2}{(K-1) \times (1+v)}$ | $F^{**} < F < F^* \rightarrow \Pi_T < \Pi_N$ | No diversification |
| Scenario 3 | $\zeta^* = 1 - \frac{N-(1-\alpha) \times \frac{\left(1+\frac{\theta}{2}\right)^{\varepsilon} \times \left(\frac{\theta}{2}\right) \times K}{(\beta-1-v)}}{K \times \left(1-\left(\frac{1+\frac{\theta}{2}}{1+\theta}\right)^{1+\varepsilon}\right)} < \zeta$ | $\zeta^* < \zeta \rightarrow N_{T,max} < N_{T2}^*$ | Sub-optimal diversification |
| Scenario 4 | $\zeta < \zeta^* = 1 - \frac{N-(1-\alpha) \times \frac{\left(1+\frac{\theta}{2}\right)^{\varepsilon} \times \left(\frac{\theta}{2}\right) \times K}{(\beta-1-v)}}{K \times \left(1-\left(\frac{1+\frac{\theta}{2}}{1+\theta}\right)^{1+\varepsilon}\right)}$ | $\zeta < \zeta^* \rightarrow N_{T2}^* < N_{T,max}$ | Optimal diversification |

*3.2. Findings of the Model*

It has been shown that there are four different scenarios to emerge in a rentier economy with absolute political authority. Any of these four scenarios will emerge depending on the dynamics of following the two critical parameters:

- The fixed cost of tradable sectors (*F*) can be relatively high or low for private agents to make a profit.
- The ruler's controlling share in the economy ($\zeta$) can be relatively high or low, which could restrict the economic diversification potential towards tradable sectors.

In the first scenario where the fixed cost of tradable sectors (*F*) is assumed too high to make any profit in the first period (poverty trap), a decrease in revenue from natural resources (*R* to *R*/2) will make tradable sectors even less attractive. Overall demand in the economy and for the tradable sectors will fall in the second period, due to falling revenue from natural resources, which will make an investment into domestic tradable sectors even less attractive. There will be no diversification into tradable sectors. Consequently, the ruler does not need to take any measure regarding the number of agents in both sectors since its share in the economy will not be compromised, as shown in Section 3. As long as the ruler controls the revenue from the natural resources (*R*/2 for the second period) along with a minimum share in nontradable sectors ($\alpha_{min}$) set in the first period, he will keep controlling more than the desired share of the economy in the second period ($\zeta Y_2$). His share of the total economy in the second period will even be higher compared to the first period under no domestic production

of tradable goods and services ($N_{T2} = 0$) with an exogenous shock of falling revenues ($R/2$). In this scenario, the economy is completely dependent on revenue from natural resources ($R$) for value addition and source of foreign income to purchase tradable goods from abroad.

In the second scenario, the fixed cost of tradable sectors ($F$) is assumed to be lower than the threshold fixed cost ($F^*$) such that even a single agent operating in that sector would make a profit. In this scenario, there will be no agents in tradable sectors at second period, due to higher profits for agents in nontradable sectors since fixed cost ($F$) is not too small to induce such a shift ($F^{**} < F$). All agents will continue to operate in more profitable nontradable sectors even though their profits decreased compared to the first period. The ruler will face a similar situation as the first case where its minimum share in nontradable from the first period ($\alpha_{min}$) was sufficient enough for him to control at least a minimum share of the economy in the second period ($\zeta$). Therefore, the ruler does not need to take any decision regarding the number of agents in nontradable and tradable sectors.

In the third scenario (scenario 3), profits in the tradable sectors will be equal to profits in nontradable sectors at some point, as shown in Figure 7. Some agents will shift their investment into tradable sectors in the second period. However, the equilibrium distribution of tradable agents ($N_{T2}$) may become larger than the maximum number of tradable agents that the ruler can tolerate ($N_{T2,max}$). In that case, the ruler will restrict entry to the tradable sectors of private agents at $N_{T2,max}$, as shown in Figure 7. The chances of entry barriers exertion by the ruler will increase when either the number of total agents in the economy ($N$) is high, or there are very high increasing returns to scale in tradable sectors compared to the wage premium ($\beta - 1 - v$), shown in inequality (25). Consequently, the economy will be below its ideal scenario, due to low diversification into tradable sectors.

In the fourth scenario (scenario 4), some private agents will shift their investment into tradable sectors until agents in nontradable and tradable sectors will have equal profits. At the equilibrium distribution, a number of tradable agents ($N_{T2}$) may be less than the maximum number of tradable agents that the ruler can tolerate ($N_{T2,max}$). This condition will hold when either the number of total agents in the economy ($N$) is relatively low, or the increasing returns to scale in tradable sectors compared to the wage premium ($\beta - 1 - v$) is not very high, as shown in inequality (26). In that case, the ruler can see a benefit of increasing entry to the tradable sectors by private agents as long as it stays below $N_{T2,max}$. The increasing number of agents into tradable sectors will increase the size of the economy and profit of the ruler while creating premium-paying jobs for the citizens. To reach that scenario ($N_{T2,max}, N_{N2}^{**}$), the ruler can increase the number of agents for the second period to lower the profit of agents in nontradable sectors to induce them into tradable sectors. This decision of the ruler will result in a higher number of tradable and nontradable agents and a larger economy, shown in Figure 10.

Finally, the ruler needs to decide on public employment ($L_{G2}$) and wages ($W_2$) for the second period in order to balance the budget. Either the ruler will discontinue half of the employees ($L_{G2} = L_{G1}/2$) or reduce wages in half ($W_1 = W_2/2$) or choose a policy in between, which may change in each scenario. At the first and second scenarios, there are no premium-paying job opportunities for citizens in nontradable sectors ($N_{T2} = 0$), which could create very high unemployment. The ruler may instead decrease the public wages ($W_2$) rather than the employment in the public sector in order to keep the rentier agreement intact (This is more or less the case in Saudi Arabia as they were able to introduce some level of the tradable sector into their economy after about 50 years of dependence on the nontradable sector compared to Qatar, for example). However, premium-wage ($1 + v$) jobs in tradable sectors are created in the third scenario, which can be filled by the citizens. The citizens would like to attain the required skills and seek jobs in tradable sectors provided that wages in this sector are higher than the wages in the public sector ($W_{G2} < W_{T2}$). The fourth scenario increases premium-wage jobs even further and hence lowers the negative effects of declining rent revenues on the employment of the citizens further.

**Table 7.** The summary of all potential scenarios.

| Scenario 1: Poverty Trap in which Tradable Sectors is not Profitable at all | | |
|---|---|---|
| **Ruler's Decision** | **Size of the Economy** | **Public Employment and Wages** |
| $N_{N2}$: No action needed $\quad N_{T2}$: No action needed | $Y_2 = (1 + (1 + \theta/2)^{1+\varepsilon}) \times R/2$ | $L_{G1} \geq L_{G2}$ $W_{G1} = 1 + u,\ (1 + u)/2 \leq W_{G2} \leq 1 + u$ |
| **Scenario 2: Tradable sectors Profitable even with a Single Agent Operating** | | |
| **Ruler's Decision** | **Size of the Economy** | **Public Employment and Wages** |
| $N_{N2}$: No action needed $\quad N_{T2}$: No action needed | $Y_2 = (1 + (1 + \theta/2)^{1+\varepsilon}) \times R/2$ | $L_{G1} \geq L_{G2}$ $W_{G1} = 1 + u,\ (1 + u)/2 \leq W_{G2} \leq 1 + u$ |
| **Scenario 3: Profits of Tradable and Nontradable Sectors Equate at a Point Higher than the Ruler can Tolerate** | | |
| **Ruler's Decision** | **Size of the Economy** | **Public Employment and Wages** |
| $N_{N2}$: No action needed $\quad N_{T2}$: Restricted to $N_{T2,max}$ | $Y_2 = (1 + (1 + \theta/2)^{1+\varepsilon}) \left( \frac{K}{K - N_{T,\ max}} \right) \times$ $R/2 < (1 + (1 + \theta/2)^{1+\varepsilon}) \left( \frac{K}{K - N_{T2}^*} \right) \times R/2$ | $L_{G1} \geq L_{G2}$ $W_{G1} = 1 + u,\ (1 + u)/2 \leq W_{G2} \leq 1 + u$ |
| **Scenario 4: Profits of Tradable and Nontradable Sectors Equate at a Point Lower than the Ruler can Tolerate in the Second Period** | | |
| **Ruler's Decision** | **Size of the Economy** | **Public Employment and Wages** |
| $N_{N2}$: Increased to $N_{N2}**$ from $N_{N2}*$ $\quad N_{T2}$: Increases to $N_{T2,max}$ from $N_{T2}*$ at scenario | $Y_2 = (1 + (1 + \theta/2)^{1+\varepsilon}) \left( \frac{K}{K - N_{T,\ max}} \right) \times$ $R/2 > (1 + (1 + \theta/2)^{1+\varepsilon}) \left( \frac{K}{K - N_{T2}^*} \right) \times R/2$ | $L_{G1} \geq L_{G2}$ $W_{G1} = 1 + u,\ (1 + u)/2 \leq W_{G2} \leq 1 + u$ |

## 4. Conclusions

We revisited The Theory of Institutions, the Rentier State Theory, Dutch Disease and the Big Push Theory for industrialization in constructing our own model, due to their partial relevance to rentier states of the GCC. The Rentier State Theory describes the economic and political nature of those autocratic and natural resource-rich states by studying the rentier agreement between the citizens and the ruler. The Theory of Institutions argues that inclusive economic institutions are key for the successful economic diversification in a developing country as it analyzes whether inclusive economic institutions can emerge in autocratic political systems. Dutch Disease Theory focuses on economic dynamics and resource allocation. The Big Push Theory discusses how developing nations can establish domestic tradable sectors and avoid the poverty trap.

Our model indicates that there are four different scenarios to emerge in a rentier economy even with absolute political authority, contrary to The Theory of Institutions. One of these scenarios will emerge depending on the dynamics of the following factors:

- The fixed cost of tradable sectors (F) can be relatively high or low for private agents to make a profit;
- The ruler's controlling share in the economy ($\zeta$) can be relatively high or low which could restrict the economic diversification potential towards tradable sectors.

Table 7 summarizes all scenarios and the potential decisions of the ruler. For the first scenario, tradable sectors are not profitable, due to relatively high upfront investment costs in this sector known as "poverty trap" [35]. Any rentier state in such a condition would continue depending on the rent revenue. The declining rent revenues would result in increasing the controlling share of the ruler. Therefore, the ruler wouldn't take any restricting action for both the tradable and nontradable sectors. The size of the economy would get smaller proportional to the decline in rent revenues (*R*).

Once tradable sectors are profitable, there emerge three potential scenarios. In the second scenario, the profits in nontradable sectors very high such that tradable sectors are not desired by the private agents. This scenario then would result in a similar situation to the first scenario. Any rentier state

in such a condition would continue depending on the rent revenue with very limited economic diversification no matter how much incentives there are. The size of the economy would get smaller proportional to the decline in rent revenues ($R$). The private agents would prefer the nontradable sectors as long as the ruler allows them business licenses to operate.

If profits in tradable sectors are sufficient enough compared to the profits of nontradable, some agents will prefer to switch into this sector. However, the ruler may allow for only a certain number of agents to invest in tradable sectors to protect his controlling share in the economy ($\zeta$). The maximum number of agents allowed to invest into tradable sectors for the second period ($N_{T2,max}$) is found as:

$$N_{T2,\,max} = K \times (1 - \zeta) \times \left( 1 - \left( \frac{1 + \frac{\theta}{2}}{1 + \theta} \right)^{1+\varepsilon} \right) \tag{27}$$

If the optimal number of agents in tradable sectors exceeds the maximum number of agents that the ruler can tolerate ($N_{T2,max} < N_{T2}{}^{*}$), then the ruler will restrict the number of agents in tradable sectors to $N_{T2,max}$. This will result in below optimal economic diversification (the third scenario, 3), a smaller economy and higher unemployment among the citizens, as shown in Figure 8. If the optimal number of agents in tradable sectors is below the maximum number of agents that the ruler can tolerate ($N_{T2,max} > N_{T2}{}^{*}$), then the ruler will put no restriction on the number of agents in tradable sectors to $N_{T2,\,max}$. In the fourth scenario (4), the ruler can increase the number of agents for the second period in order to attain some higher number of tradable agents to maximize economic diversification. The increasing number of agents will decrease the profits of agents in nontradable sectors. This will prompt some of those agents to invest in tradable sectors as long as $N_{T2}{}^{*}$ doesn't exceed $N_{T2,max}$ of the ruler's tolerance limit, as shown in Figure 9. In the fourth and last scenario, the economy will attain an optimal diversification within the same political structure.

The desired economic diversification outcome for any rentier state is the fourth and last scenario where the country achieves an optimal economic diversification. In that scenario, the government (ruler) controls a significant share in the non-tradables sector such that the growing role of private agents in the tradables sector is not a threat to its political authority. Many governments in the rentier states have actually significant controlling shares in the non-tradables sector, such as telecommunication, healthcare, transportation and education services [6,10]. Therefore, most of these governments actively encourage economic diversification to tradables sectors by the private agents instead of opposing to it since that diversification will also serve to its benefit in the form of new employment opportunities created for the citizens and lessened reliance on the rent revenues [21]. However, a successful economic diversification into the tradables sector by the private agents requires incentives in terms of higher profitability compared to the non-tradables sector. The model developed here suggests that this can happen only in two ways: Low profitability prospects in the non-tradables sector or high profitability prospects in the tradables sector through higher productivity level. To decrease the profitability level in non-tradable sectors, the government may stop giving exclusive licenses to business families and create a more competitive and equal playing field. This may hurt the interests of those families and may jeopardize the rentier agreement between them and the government [44]. The second option is to increase the profitability prospects in the tradables sector through higher productivity level by investing in education, physical infrastructure and establishing business incubation centers, as well as channeling cheap credit to such sectors. Many rentier states are reluctant to follow the first option to avoid altering the social contract with some of its citizens [45]. However, there is a significant level of investment into the physical infrastructure, education, opening of free zones, business incubation centers and all other forms of support in a bid to push and attract investment into the tradables sector in rentier states [12,45].

## 5. Discussions

Considering the fact that many resource rich countries are facing civil strife and chaos, such as Libya, Venezuela, and Syria, the issue of effective economic diversification has become urgent more than ever. Among the myriad of reasons for the necessity of effective economic diversification in rentier states, effects of climate change, decreasing role of fossil fuels in global energy and transportation systems and rapidly increasing youth population have become critical issues for sustainable development in those rentier states. To our knowledge, the proposed model is the first ever for an effective economic diversification in a rentier state to sustain the welfare of the citizens and the political status-quo. The model borrows a significant number of its assumptions from the development of economics theories, such as Dutch Disease Theory, The Theory of Institutions and The Big Push Theory. The model explores the potential of economic diversification towards tradable (non-rentier) sectors in a rentier state. The benefits of economic diversification for the ruler will be less pressure on government budget and the potential windfall from a growing economy, due to its share in nontradable sectors. A successful economic diversification can create premium wages for the citizens and encourage them to seek a job in the private sectors (non-rentier) which will lessen the pressure on the government. A rentier government will be supportive of diversifying its economy towards tradable sectors as long as the economic power of itself (the ruler) is not compromised. The ruler of a rentier state who has sufficient economic power may lessen or remove the entry barriers for the private businesses and hence stimulate economic diversification further. On the other hand, a ruler who does not have sufficient share in the economy will react to declining rent revenues by exerting more entry barriers or resort to grabbing of private property. Both of these actions will cause economic institutions to become more extractive and hence stall the economic development of the country as argued by The Theory of Institutions [29].

The model developed in this paper is transitionary where the effect of savings by the ruler, the private agents, and the citizens are ignored. Similarly, the model keeps the important role of the rent revenue ($R$) for the economy albeit of its decreasing value. Any future work should take these two factors into account in order to envisage a complete diversification away from the rent revenues. Furthermore, capital movement across the borders will complete the analytical model proposed in this paper. The relationship between the price of nontradables and rent revenue is assumed to be exogenous following The Dutch Disease Theory [37]. The future work in which this relationship endogenized can give a better understanding of economic diversification dynamics in rentier states. Most rentier states have invested their excess rent revenues to the overseas investment vehicles during the times of high prices for those commodities, such as oil and gas [46]. Return from those investments in the future may play a cushion for decreasing rent revenues which this paper didn't take into account as well. Sovereign wealth fund practices of rentier states and the effects on the domestic economy when rent revenues decline is an interesting future research topic [47]. Many rentier governments have invested into the tradables sector, such as the aluminum and steel industries [22]. Therefore, the tolerance level of the government for private agents in the tradables sector can be higher than what predicted in the model as long as the interests of these two entities don't collide.

**Author Contributions:** A.K.: writing, methodology, formal analysis validation; A.K., E.T., T.M. and M.K.: investigation, original draft preparation; A.K., E.T. and I.-T.T.: writing—review and editing.

**Funding:** This research was funded by the Qatar National Library.

**Acknowledgments:** We acknowledge the financial support provided by both Masdar Institute of Science and Technology (part of Khalifa University) and Hamad Bin Khalifa University (HBKU). A sizeable part of this paper has also been published in the lead author's doctoral thesis.

**Conflicts of Interest:** The authors declare no conflict of interest.

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
