# Peer review of "Economic Diversification Potential in the Rentier States towards a Sustainable Development: A Theoretical Model"

_sustainability, doi:10.3390/su11030911_

Round 1
Reviewer 1 Report
First of all, I would like to congratulate the authors for the paper, which is very interesting and whose theme is related to the Sustainability journal.
The structure of the paper is adequate, although, to adapt it to the standards of the journal and to make it clearer to readers, it would be convenient to differentiate, visibly, the sections on methodology, results and discussion.
I would like to make some recommendations that could increase the quality of the paper presented.
- Arabian regions are presented as an example of rentier states; however, there are currently countries like these in America or Africa, so it would be appropriate to quote some of them in this paper.
- In the Introduction, a hypothesis must be clearly presented, which will be verified with the objectives set out at the end of page 2, as well as the methodology in a more general way.
- In the Introduction, on line 74 on page 2, the authors affirm that the current literature on rentier states tends to ...; this should be supported, precisely, by citing that literature.
- The authors must support with scientific literature the statements they make from lines 81 to 87, or do they correspond to their hypothesis?
- The ideas on which the theoretical model is based are suitably presented, as well as the variables chosen and the scenarios (four are observed in the model). However, I think that it should be explained what a theoretical model consists of and what its benefits are scientifically in order to justify the fact of constructing a theoretical model.
Author Response
We thank to the 1st reviewer for the constructive and critical comments. We have been through all of these comments and rewritten the related parts of the paper by editing the existing structure and adding further support from the literature and our own to strengthen the argument of the paper. The changes are highlighted with yellow in the new version of the paper and we also give specific location of each change to each point of the reviewer below.
· Other rentier states are also mentioned in lines from 51 to 53.
· Lines from 54 to 119 are edited in order to explain first rentier states need urgent economic diversification which have been lackluster so far. Then we explained why a theoretical model was developed in order to find an effective economic diversification and hence a sustainable development path to the future.
· We cite two papers regarding the fact that political system or dynamics of rentier states generally overlooked in economic diversification studies from line 86 to 119.
· We add further support from the literature to our assumptions in lines from 99 to 102.
· We also explained clearly that the proposed model may deliver policy suggestions for an effective economic diversification and hence a sustainable development in terms of preserving the political stability and economic prosperity even when the rent revenues go down from in lines from 125 to 164.
Reviewer 2 Report
I appreciate that the authors havean approach to an important topic - theoretical model to analyze whether a rentier state can diversify its economy away from the rent revenue...
Research methodology seems to be fine.
Some references - papers from top magazines - appeared between 2016 - 2017- and 2018. The bibliography should be strengthened with this type of articles.
The conclusions should be the authors' own. Conclusions may also need to be reviewed (possibly shortened).
Author Response
We thank to the 2nd reviewer for the constructive and critical comments. We have been through all of these comments and rewritten the related parts of the paper by editing the existing structure and adding further support from the literature and our own to strengthen the argument of the paper. The changes are highlighted with yellow in the new version of the paper and we also give specific location of each change to each point of the reviewer below.
· Based on the suggestion of the reviewer, we added new references numbered from 24 to 27 which published in the last two years regarding the sustainable economic development of rentier states
· There are also other references such reference 12, 16 or 17 which are either published in 2016 or 2017
· Following the advice of all the reviewers, we have separated the conclusions from the discussions as we put later into another section. We also shortened the conclusion part as much as we can do.
Reviewer 3 Report
The article deals with a topic of potential theoretical scientific interest but is not well focused. The manuscript is too long.
Some comments in order to improve the scientific soundness of the paper would be:
-In a journal such as Sustainability one expect that authors address some topic related with the sustainiability of destinations or theorethical model applied. In this paper there is no clear references with the sustainaiblity issues of the analyzed model. Please include some issues on this topic affecting models both in introduction also in conclusions/discussion section.
-The literature review is giving one paragraph information but not clear on what this paper is focusing on; without going into the most relevant aspects that would justify the theoretical model proposed. I would like to see a more critical adoption of the literature being used. ok lines 71-78 to detail the objectives of the study.
-lines 137-139: why? just at the end of introduction (lines 94-96) the authors explained the structure of paper.
-3. and 3.1 Paragraphs lines 359-708: these paragraphs in your current form are difficult to follow. Probably it's better to insert an appendix or to summarize all.
I suggest to make a paragraph with Discussion and a Conclusion. It is innovative the use of this model for international cases or there has been previous studies using the same models?
The former should present practical implications of this study and comparison to the other similar studies, whereas the latter should bear the only conclusions (please, provide several conclusions derived from 3.2 findind of the model lines 709-776) and perspectives for further research.
Highlight implications for sustainability of the modeI .
Author Response
We thank to the 3rd reviewer for the constructive and critical comments. We have been through all of these comments and rewritten the related parts of the paper by editing the existing structure and adding further support from the literature and our own to strengthen the argument of the paper. The changes are highlighted with yellow in the new version of the paper and we also give specific location of each change to each point of the reviewer below.
· We discuss the future of sustainable development in rentier states from mainly economic diversification that may result in decreased dependence on rent revenue and job creation for the citizens. From line 54 to 119, we discuss the challenges faced by the rentier states regarding a more sustainable economic development and whether an economic diversification is possible within the current political system.
· We particularly focus on the necessity and importance of a theoretical model in assessing the economic potential of a rentier state for two important reasons. Firstly, the political change is a risky and chaotic process and thus we assume the current political system as unchanging in those states. We delve later on to the potential of economic diversification in those states under the existing system by developing a theoretical model which has not been done before in the rentier state literature.
· Lines from 137 to 139 were removed as suggested by the reviewer to avoid repetition.
· We have rewritten from 125 to 164 to summarize what are the steps in solving the theoretical model and its relevance to the sustainable development in rentier states.
· We separated the conclusions and discussions while rewriting the both sections to highlight the important findings of the model and uniqueness and connection of the model to the rentier state and development economics literatures.
· We discuss the shortcomings and future work on the model at the discussions section 874 to 911.
Round 2
Reviewer 3 Report
The manuscript has technical merits in terms of its contributions. Now it is improved.
Line 802 Conclusion: probably it's discussion (line 873). Usually conclusions are at the end of the article.